# Quality of life and multiple long-term conditions in Southeast Asia: a systematic review and meta-analysis

Deborah Ikhile [1,2,9] ✉, Patrick Highton [1,2,9], Clare Gillies[1,2], Ruksar Abdala[1,2], Ashkon Ardavani [1,2], Monika Arora [3,4], Amrit Banstola [5], Aakrushi Brahmbhatt[6], Shabana Cassambai [1,2], Mark P. Funnell [1,2], Shifalika Goenka [6], Shavez Jeffers [1,2], Dimple Kondal[6], Sailesh Mohan[6], Prakash Mulakalapalli [6], Natalia Oli[7], Arron Peace [1,2], Kuldeep Singh [8], Abhinav Vaidya[7], Nikhil Srinivasapura Venkateshmurthy [6], Dorairaj Prabhakaran [6,10] & Kamlesh Khunti[1,2,10]

This review systematically synthesised the evidence on quality of life measures and outcomes for people living with multiple long-term conditions in the Southeast Asia region. Results were analysed using a combination of methods, meta-analysis for studies where the same quality of life score was reported across three or more cohorts, and descriptive narrative synthesis. In total, 34 studies comprising 11,876 participants were included in the narrative synthesis and 14 of these were included in meta-analysis. The most common quality of life tools used included WHOQOL-BREF ($n = 8$) and EQ-5D-5L ($n = 3$) with pooled mean values of 70.47 (95% CI: 62.71 to 78.24) and 0.76 (95% CI: 0.67 to 0.84) respectively, indicating reduced but good quality of life. As healthcare systems adapt to the evolving challenges associated with multiple long-term conditions, understanding the tools and measures used to assess quality of life in different contexts becomes imperative to account for disease combinations and cultural nuances.

The increasing prevalence of multiple long-term conditions (MLTC) poses a significant challenge to patients and healthcare systems globally[1]. MLTC or multimorbidity, defined as the concurrent existence of two or more chronic conditions[2], has an estimated global prevalence of ~37%[3]. Though, this varies across regions and within countries due to differences in definitions of MLTC, age and in social determinants of health[3]. For example, a recent systematic review reported the highest MLTC prevalence in community settings in South America at 45.70%, compared to 35% in Asia[3]. Even within Southeast Asia, notable regional variations have been reported, ranging from 8.40% in Bangladesh[4] to ~22% in India[5] and 25.10% in Nepal[6]. The prevalence of MLTC also varies with age. A nationwide cross-sectional study in India, reported a prevalence of 7.20% among individuals aged 15–49 years[7], while a longitudinal ageing study reported a prevalence exceeding 30% in adults aged 60 years and above[8]. It is crucial to note that estimated burden of MLTC in the region, as is the case in other Low-and-Middle-income Countries (LMICs) is likely underestimated due to suboptimal reporting and poorly integrated health systems[2]. With global populations aging, the number of people living with MLTC is expected to increase in the coming years[9].

[1]Diabetes Research Centre, University of Leicester, Leicester, UK. [2]National Institute for Health and Social Care Research Applied Research Collaboration East Midlands, Leicester, East Midlands, UK. [3]Health-Related Information Dissemination Among Youth, New Delhi, India. [4]Public Health Foundation of India, New Delhi, India. [5]Department of Health Sciences, Brunel University of London, Middlesex, UK. [6]Centre for Chronic Disease Control, New Delhi, India. [7]Kathmandu Medical College, Kathmandu, Nepal. [8]All India Institute of Medical Sciences Jodhpur, Jodhpur, India. [9]These authors contributed equally: Deborah Ikhile, Patrick Highton. [10]These authors jointly supervised this work: Dorairaj Prabhakaran, Kamlesh Khunti. ✉e-mail: di46@leicester.ac.uk

The rising burden of MLTC is associated with significant individual, healthcare, and economic challenges, including the complexity in patient care, reduced quality of life (QOL), higher mortality rates, increased healthcare utilisation, and substantial economic costs such as greater Gross Domestic Product (GDP) spending[3,10]. In LMICs such as countries in Southeast Asia, factors such as increasing urbanisation, the persistent burden of infectious diseases, higher MLTC prevalence at an earlier age and healthcare systems primarily oriented towards the management of single conditions complicate access to care for people with MLTC[11,12]. These subsequently adversely affect MLTC outcomes, including mortality, hospital admission rates, disability rates, and patient reported outcomes such as QOL[13]. A multidisciplinary collaboration recently identified health related QOL as a core outcome for MLTC intervention in LMICs[14]. Other studies similarly highlight QOL as a core outcome of importance, particularly as people with MLTC often experience a decline in both physical and mental health, leading to disability and decreased QOL, and mortality rate alone may not adequately reflect the impact of these conditions[15]. This is the case in Southeast Asia where QOL has been identified as an important patient reported outcome for MLTC patients particularly for those routinely seen in primary care settings[16].

A systematic review conducted by Haraldstad et al.[17], of QOL research in medicine and health sciences identified a variety of questionnaires used to measure QOL, both generic and disease-specific. The most common generic measures included the Short Form-36 (SF-36), EuroQOL 5D (EQ-5D), World Health Organisation QOL Brief version (WHOQOL-BREF), and SF-12[17]. Although disease-specific tools have been developed to assess QOL associated with single long-term conditions like the Parkinson's Disease Quality of Life Questionnaire[18], the assessment of MLTC is complicated due to the complex clusters and interactions of conditions. Overall, QOL tools gather subjective information about individuals' well-being across physical, psychological, and social dimensions of QOL, all of which can be profoundly affected by MLTC[13].

While several systematic reviews have been conducted on QOL and MLTC[10,13], there remains a lack of comprehensive synthesis of primary studies relating to the Southeast Asia region. Synthesising evidence on QOL is essential for healthcare practitioners, policymakers, and researchers to inform the development and refinement of QOL tools and measures tailored to the intricacies of managing people with MLTC in Southeast Asia. Therefore, in this review, we synthesise and critically analyse the available literature on the tools and measures used to assess QOL in people living with MLTC in Southeast Asia and their outcomes. By focusing on QOL, this systematic review aims to contribute to the ongoing global efforts in optimising healthcare strategies and improve outcomes for individuals with MLTC.

## Results

A total of 52,614 titles were collectively identified through the database searches. Following the removal of duplicate publications, 33,664 studies were screened during the title and abstract stage, which resulted in the exclusion of 31,818 irrelevant studies. The full texts of the remaining 1846 studies were assessed for eligibility, resulting in 34 studies included in this review (Fig. 1).

### Study characteristics

A total of 11,876 participants were included in the eligible studies ($n = 34$), with study sample size ranging from 32 to 2919 participants. Most of the studies ($n = 22$) used a cross-sectional study design[19–40], seven studies used a prospective design[41–47], three were randomized control trials[48–50], one study was a quasi-experimental design[51], and one was mixed-method[52]. Mean age of the cohorts ranged from 35.4 to 68.9 years. Studies were conducted in India ($n = 17$[20,23–25,27,28,31,32,34,35,40,43–47,52]), Thailand ($n = 9$[21,30,33,37,38,41,42,48,50]), Nepal ($n = 3$[22,26,29]), Indonesia ($n = 3$[19,49,51]), Bangladesh ($n = 1$[36]), and Sri Lanka ($n = 1$[39]). The overall

quality rating of included studies was good ($n = 10$[19,24,25,27,32,41,43,46,48,50]), fair ($n = 22$[20–22,25,28–34,36–38,40–45,47,51]), or poor ($n = 2$[49,52]). A sensitivity analysis excluding one of these studies resulted in a lower pooled EQ-VAS mean score of 58.43 (56.35, 60.51)[49]. The second poorly rated study utilised the Parkinson's disease questionnaire[52], but as this was the only study using this tool a sensitivity analysis for study quality could not be carried out. Study and participant characteristics of the included studies are detailed in Supplementary Table 1.

### Disease combination

There was notable variation in the disease combination among the study participants, and conditions were combined into four categories as presented (Supplementary Table 2). Nine studies reported a prevalence of 100% for the two eligible conditions among the MLTC sample populations[22,23,35,40,43,46–49], others were based on a combination of one indexed condition (100%) and another eligible condition with prevalence ranging from 70.0–88.5%. Only two studies reported a combination of three conditions[24,52], all other studies reported a combination of only two conditions.

### Quality of life tools

The most common QOL tools used were WHOQOL-BREF ($n = 8$[20,27,29,31,32,41,42,48]), EQ-5D-5L ($n = 3$[19,22,33]), EQ-5D-3L ($n = 3$[23,24,39]), EQ-VAS ($n = 2$[22,49]), and SF-36 ($n = 2$[40,46]). These tools were generic and not designed for a specific single disease. Some tools were combined and used to assess general and specific health measures, such as the kidney disease quality of life short form (KDQOL-SF)[21,25,45,51] and Kidney Disease Quality of Life 36 item questionnaire (KDQOL-36)[44]. Specific tools designed for the indexed conditions used to measure QOL included Parkinson's Disease Questionnaire-8 (PDQ-8)[52], WHOQOL-8[26], Appraisal of Diabetes Scale (ADS)[27], The Diabetes-39 Questionnaire(D-39)[37], Audit of Diabetes Dependent Quality of Life (ADDQOL)[38], Dhingra and Rajpal-12 scale (DR-12)[43], Stroke-specific QOL Scale (SS-QOL)[47] Breast cancer specific EORTC QLQ[28], Functional Assessment of Cancer Therapy-General (FACT-G) scale[36], Seattle Angina Questionnaire (SAQ)[46], St George's Respiratory Questionnaire (SQRQ)[34], Glaucoma QOL Questionnaire (GQL-15)[35] and the 9-item Thai Health Status Assessment Instrument (9-THAI)[30]. It was unclear whether the Pictorial Thai QOL Test was generic or disease-specific[50]. Two studies combined two different scales: WHOQOL-BREF + ADS[27], and SF-36 + SAQ[46]. Twelve studies reported that the QOL tools used were locally validated or translated[19,21,25–27,29,31–33,38,41,50]. Two studies used locally developed QOL tools[30,43]. None of the tools in the included studies were developed specifically for MLTC. The generic tools measure overlapping constructs broadly categorised into four core domains of physical, mental, social and global wellbeing constructs (Supplementary Fig. 1). An additional domain covering disease or symptom specific components is included for the five studies using disease-specific tools.

### Quality of life outcomes

Out of the 34 studies included, only 14[19,22,27,29,33,34,36,39,43,48–52] qualified for inclusion in the meta-analyses (Fig. 2). The reported mean QOL scores for individuals with MLTC in the studies identified are depicted in Fig. 2. Across 14 studies, 8 QOL tools had been used. For the meta-analyses where more than three studies reported the same tool, the pooled mean value for EQ-5D across 5 cohorts was 0·76 (95% CI: 0.67 to 0.84) and for WHOQOL-BREF across four cohorts it was 70.47 (62.71 to 78.24), indicating perceived good QOL. For both meta-analyses, between study heterogeneity was high, $I^2$ was 98·7% and 98·1% respectively. Across the 14 studies reporting overall mean for QOL, 11 rated the scores as good ($n = 5$[19,20,22,27,33]), average ($n = 3$[27,29,39]) or poor ($n = 3$[34,36,52]), based on indices used in the primary studies (Supplementary Table 1). Some studies reported QOL data by domains and this varied across, and within tools. Pain or discomfort was the most common domain reported when using the EQ-5D tools. For instance, in

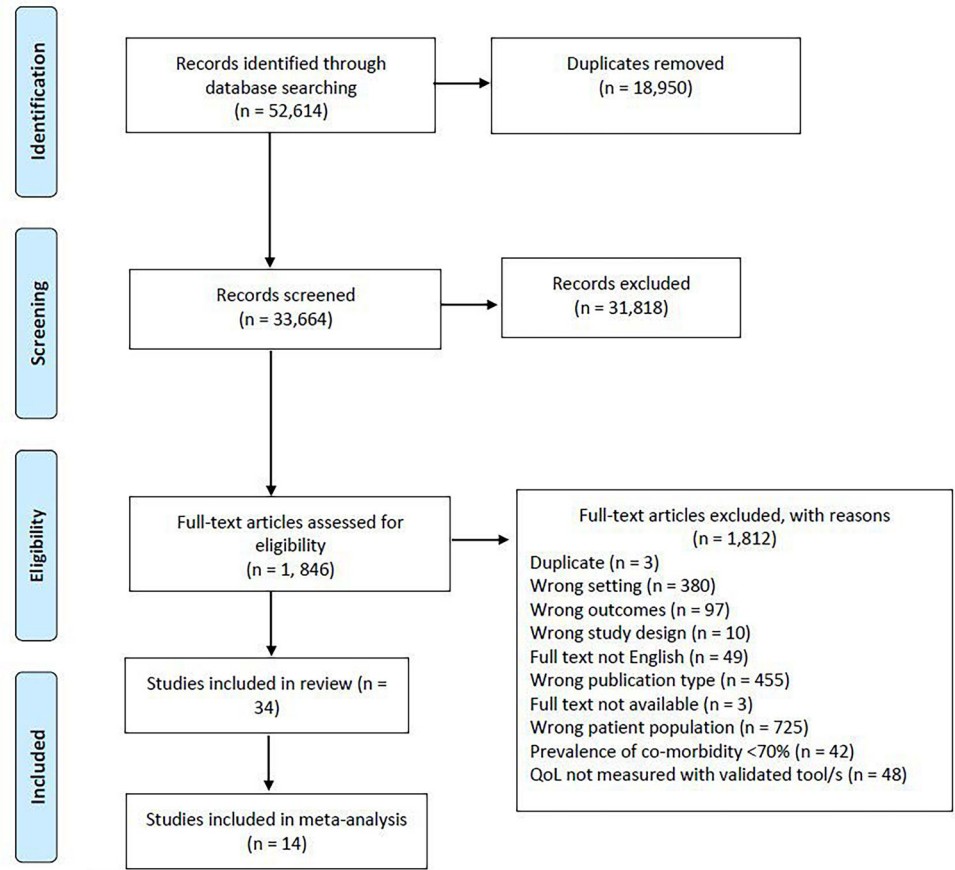

**Fig. 1 | PRISMA flow chart.** This figure shows the systematic process we followed to select the papers included in our review. From the 52,614 titles captured by our database search, we screened 33,664 and included 34 studies in the review.

the Alfian et al. study conducted in Indonesia, 47.4% of the MLTC population reported pain or discomfort[19]. Similarly, the physical health component was the most affected when using the KDQOL[45] and WHOQOL-BREF tools[27]. Whereas anxiety/depression was the most reported component among all domains of the EQ-5D-3L in one study[23]. Another study using WHOQOL-BREF reported the lowest QOL for the social domain with the highest observed in the environmental domain[20]. However, QOL scores were better in the social domain in the Patel et al. study[27]. For prospective studies, there were significant improvements in QOL outcome between baseline and specified time points post-treatment[42,43,46,50,51] except in one study, Fadhil et al., which showed no significant difference in QOL outcome from baseline to the end of the study at 8 weeks[48]. Another study conducted in Indonesia showed significant improvement in QOL from $47.25 \pm 7.29$ at baseline to $61.20 \pm 5.80$ three weeks post-intervention.

### Factors associated with quality of life

**Socioeconomic factors.** A total of 14 studies highlighted socio-economic factors associated with QOL[20–23,26,27,29,34,36,37,40,44,47,52]. Female gender[27,34,44,47,52], older age (>65 years)[21,22,47], lower education/literacy level[20,22,40,44], being unemployed[23,40], engagement in hazardous occupations[44], being a non-professional[44], lower socioeconomic con-dition/income level[26,40,44], rural residence[44], and non-vegetarian dietary pattern[44] were all associated with lower QOL. Certain ethnicities/castes were also associated with lower QOL[26]. For instance, in one study, non-Mongolian (Brahmin, Chettri, Madheshi) participants were identified to have lower QOL than Mongolian participants[26]. Similarly, another study reported that Brahmin/Chhetri and other ethnic groups had lower QOL than Aadibasi/Janajati ethnicity[29]. Employment status did not show any significant association with QOL in one study[22].

Conversely, financial independence for health care needs[23], high lit-eracy among patients[20], higher socioeconomic status and higher educational status of the study participants[21,29], being female[20], and better diabetes self-management practices[37], were associated with higher QOL scores. Although employment status showed no associa-tion with QOL on the EQ-5D-5L index, it was associated with an increased EQ VAS score[22]. Age, sex, residence and marital status were not associated with QOL using the WHOQOL-BREF tool[29]. In some instances, socioeconomic associations were presented by domains. For instance, being female was associated with lower scores in the global health domain for WHOQOL-BREF[27]. Furthermore, age and income were associated with the physical component of the KDQOL-SF[21], while age >50 years old was associated with lower scores for the mental component of KDQOL-36[44]. In people living with metastatic cancer and depression, older age and higher financial difficulty were associated with worse functional well-being[36].

**Clinical factors.** Fourteen studies showed associations between clin-ical factors and QOL[19,21,24,25,27,29,33,35,37,42–45,47]. Medication adherence[40], frequency of dialysis[21], physician-related distress[19], body mass index[44], albuminuria[44], participation in a rehabilitation programme[42], metfor-min therapy[43], Haemoglobin/haematocrit level and adequacy of dialysis[45], controlled diabetes[27], ischemic stroke compared to hae-morrhagic stroke[47] and renal transplant[29] were associated with higher QOL scores. The presence of anaemia in people with diabetes and CKD[24], CKD stages III-IV[24], emotional distress[19], interpersonal distress[19], regimen distress[19], prolonged dialysis recovery time[25], presence of cardiovascular disease[37], number of non-pharmacological measures[27], severity of glaucoma[35] and haemodialysis[29] were associated with lower QOL. For domain associations, participants with normal cognition

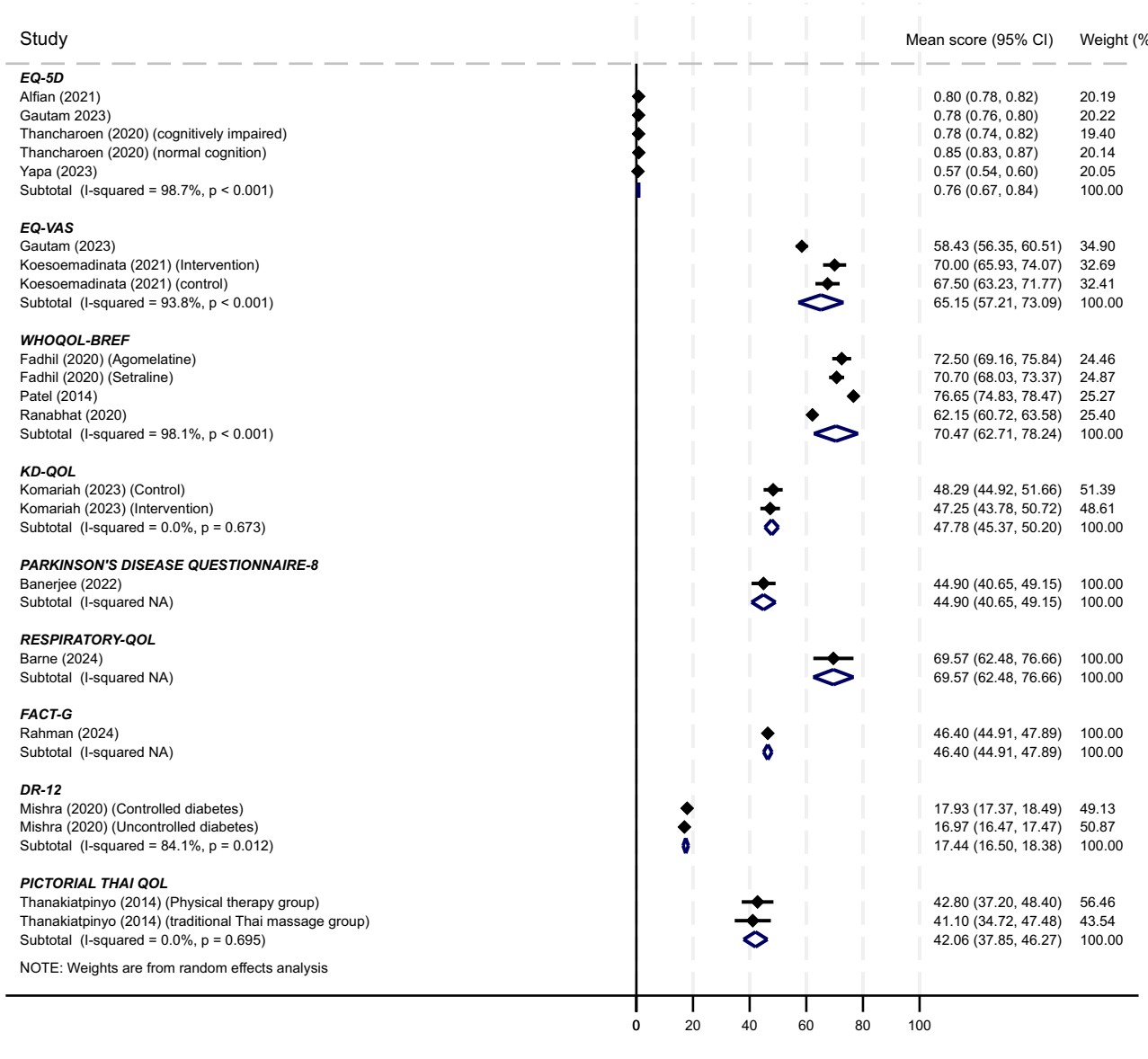

**Fig. 2 | Meta plot of QOL tools.** This figure shows the pooled mean QOL scores and associated 95% confidence level intervals for the 14 studies eligible for meta-analysis. EQ-5D-5L tool was used in all outlined studies, except Yapa (2023)[39], where EQ-5D-3L was used. Random effects meta-analysis was used to pool the QOL scores, and between study heterogeneity was quantified using the I-squared statistic. The p-values are associated with the I-Squared values for between study heterogeneity. A p-value less than 0.05 indicates statistically significant between study heterogeneity. The tests are 2-sided.

showed an association with the pain/discomfort domain of EQ-5D-5L[33], lower estimated glomerular filtration rate was associated with worse scores for the mental component of KDQOL-36[44]. Participants who used alternative therapies had better scores on the mental component summary, physical component summary, and burden scales and worse scores on the symptoms scale of KDQOL[44]. WHOQOL-BREF domain scores were higher in patients with controlled diabetes as compared to uncontrolled, except in the case of physical domain scores[27].

## Discussion

People living with MLTC experience a greater disease burden compared to those with a single chronic condition or no chronic conditions. This review identified a variety of generic and specific tools used to assess QOL among people with MLTC in the Southeast Asia region. However, no QOL tool specifically designed for MLTC was found. QOL outcomes from this review were reported as average or good for most studies and mediated by clinical and socioeconomic factors. For

instance, the mean QOL index score (where reported) for EQ-5D-5L ranged from 0.78 to 0.84, indicating an overall high QOL index in these populations.

In Southeast Asia, the most commonly used QOL tools were the WHOQOL-BREF and the 3-level and 5-level EuroQOL 5-Dimension (EQ-5D) questionnaire[19,22–24]. This contradicts recent evidence showing that short form (SF) questionnaires are the most widely used QOL tools for MLTC[10]. However, it must be noted that the review mentioned was not restricted to one specific region, whereas the current review is specific to Southeast Asia. Although only two studies in this current review used the SF-36, it is important to highlight that some specific tools were based on it. Both the general measures of the KDQOL-SF and KDQOL-36 are based on questions from the SF-36. The availability of translated and locally validated versions likely influenced the type of QOL tool used. For instance, one study included in this review used the EQ-5D-5L because of its superior measurement and scoring properties in Indonesian type 2 diabetes populations[19]. This underlines the need

for cultural adaptations of QOL measures for MLTC to enhance their relevance, acceptability, and sensitivity to the diverse cultural contexts of Southeast Asian populations.

The combination of diseases in this review aligns with the National Institute for Health and Care Research definition of MLTC, i.e., a combination of two or more physical non-communicable diseases (NCDs), physical NCD(s) and infectious disease(s), two or more infectious diseases, or a combination of a physical condition(s) and a mental health condition(s)[53]. While some studies have restricted the focus of MLTC to NCDs only[54], findings from this review emphasise the importance of exploring the influence of infectious diseases, which remain a significant health challenge in LMIC's. In Southeast Asia, tuberculosis continues to be a public health priority accounting for 45% of the global incidence in 2021[55]. Although the disease combination did not follow existing patterns for any of the eligible countries, hypertension and diabetes were the most frequently cited conditions, consistent with their status as leading chronic conditions in the region[16]. Understanding disease combination and clusters is crucial for selecting appropriate QOL tools. This review found that for patients with diabetes the EQ-5D or WHOQOL-BREF tools were used, emphasising the need to tailor QOL tools to specific disease combinations and regions.

The multidimensionality of QOL is a key consideration in developing a QOL measure for MLTC. QOL measures are inherently both subjective and multidimensional, and they typically cover three broad domains of physical, social, and psychological constructs[13]. More recent measures are expanding into other domains such as cognitive, environmental and economic[56,57]. This review demonstrated that the three key dimensions of physical, psychological and social domains remain crucial for any QOL tool assessing MLTC. However, it also highlights the importance of a fourth domain, global wellbeing, that encompasses wider constructs, such as environmental and economic domains. Additionally, this review adopts the terms "mental" rather than psychological construct to cover both emotional and cognitive aspects. Future research should explore the domains and mapping components of a bespoke QOL tool and measures for MLTC to ensure a patient-centred care for people living with MLTC.

Patient-centred care requires an in-depth understanding of factors that influence QOL. Findings from this review indicate a robust link between socioeconomic or clinical factors and QOL in patients with MLTC, as established from a recent study in India[16]. However, socioeconomic discrepancies across different countries in India need to be noted. The association between socioeconomic factors, such as older age and poor QOL, is consistent with existing evidence from high income countries, where individuals over 60 years old tend to report lower QOL[58]. This further strengthens the association between MLTC and ageing and reiterates the need for research to improve QOL for older people with MLTC[15,58]. Also, findings show variations by sex, with studies included in this review reporting higher prevalence in women[27,34,44,47,52]. This aligns with data from the Longitudinal Ageing Study in India, which included 31,464 older adults and showed that women were 1.6 times more likely than men to have MLTC[8]. This review also identified several clinical factors associated with QOL outcomes, including treatment modality, e.g., haemodialysis versus renal transplant in end-stage renal disease patients[29]. Investigating the influence of clinical and socioeconomic factors on QOL is pivotal for addressing disparities and tailoring MLTC interventions effectively.

This current review did not reveal any associations between environmental factors and QOL. More studies are required to explore this association, given the growing interest and links between climate change, sustainability issues and health[57], the region's high vulnerability to natural disasters and pollution[59]. There is a need for further studies on the social determinants of MLTC in the region, particularly focusing on gender and socioeconomic status, as these can significantly impact upon QOL outcomes. Future research should also

consider the community's role in addressing MLTC, as social domains of QOL such as stigma-related issues and informed decision-making are best tackled by appropriate community engagement and involvement. It is also crucial for healthcare professionals to understand how MLTC affects patients' QOL and how this should be properly measured, as patients with MLTC are routinely seen in primary care, such as in India[16].

Our review also has significant policy implications, particularly in relation to QOL and long-term disability management in resource-constrained healthcare systems. Although limited data exists for the Southeast Asia region, a meta-analysis published in 2024 estimated a high prevalence of disability among people with MLTC at 34.9% (95% CI = 25.8–43.9%)[60]. The effect of MLTC on long-term disability exacerbates healthcare utilisation and expenditure, increases caregiving burden, and the need for rehabilitation and longterm care, which many health systems in the region are not yet well equipped to provide. Policy responses need to include patient centric approaches rather than disease specific approaches, rather than disease specific vertical approaches, investment in universal health coverage as a way of addressing disparities due to the social determinants of health and promoting preventive efforts, particularly those targeting modifiable risk factors. Also, investment in quality curative and rehabilitation services is required to mitigate adverse impact on QOL due to disability resulting from MLTC.

This review is the first to provide robust evidence on the existing QOL measures used in people living with MLTC and their outcomes in Southeast Asia. However, the findings should be interpreted while taking account of the following limitations and considerations. Firstly, the underestimation of MLTC burden in the region leads to an incomplete assessment of QOL outcomes in people living with MLTC. Secondly, the mean value from observational studies is more likely to be representative than RCTs as RCTs will have exclusion criteria which may mean QOL values are higher than the population averages for the population of interest. One key limitation of this review was the lack of standardised definition of MLTC and absence of QOL data in the majority of the studies reviewed. To address this, a cut-off of 70% for eligible conditions outside the indexed condition was applied, providing a standardised approach to determining the MLTC eligibility in study populations. Effective assessment of the QOL outcomes was also limited by heterogeneity of study design, lack of standardisation in reporting, and missing data, contributed to the inability to undertake a meta-analysis for all included studies. The narrative synthesis undertaken in this review, underpinned by a robust evidence base, provided a systematic approach for synthesising and analysing the data, in order to provide better understanding of factors influencing QOL in MLTC patients. A comprehensive search was conducted using multiple databases. Deliberate steps were taken to minimise bias, such as including studies where MLTC status was established prior to QOL outcome assessment. For studies using the EQ-5D tool, anxiety/depression was treated as an outcome, not an eligible exposure, given the tool's measurement of these domains. Another potential limitation was the study design used in the primary research. The majority of the primary studies included were cross sectional, this remains a challenge for MLTC systematic reviews as observed in a recent review[10]. As MLTC is dynamic, cross-sectional studies may not adequately capture its impact on QOL. In order to effectively understand the changing effect of MLTC on QOL over time, longitudinal studies are required.

In conclusion, this review, which synthesised QOL measures and outcomes in people with MLTC in Southeast Asia, showed that although there is no specific QOL measure for MLTC, there is an opportunity to develop a multidimensional measure to better the understanding of MLTC on QOL. The generic and specific tools identified in this review share commonalities that cut across four key dimensions of physical, mental, social and global wellbeing. These dimensions could form the core of a future bespoke QOL measure for

MLTC. Given these findings, further mixed methods research should map domains across various QOL tools, gather perspectives and build consensus with multi-stakeholders for development of a MLTC QOL measure. However, it may be necessary to create separate tools to account for the differences in disease combinations and MLTC clusters between different populations and countries.

## Methods

### Study strategy and selection criteria

This systematic review, meta-analyses and narrative synthesis were conducted and reported in line with the Preferred Reporting Items for Systematic Reviews and Meta-Analyses (PRISMA) checklist[61] (Supplementary Table 3) and the Synthesis without meta-analysis checklist guidance (SwiM) in systematic reviews[62] (Supplementary Table 4). MEDLINE, CINAHL, Embase, Web of Science and The Cochrane Library databases were systematically searched from inception to August 2024 using the search strategy provided in Supplementary Table 5. All original articles published in English language (both qualitative and quantitative) assessing any tool and measure of QOL (as either primary or secondary outcome) in patients with MLTC residing in the Southeast Asia region, as defined by the World Health Organization South-East Asia as of August 2024[63], were included. These included Bangladesh, Bhutan, Democratic People's Republic of Korea, India, Indonesia, Maldives, Myanmar (Burma), Nepal, Sri Lanka, Thailand, and Timor-Leste (East Timor, Democratic Republic of Timor-Leste). It is important to note that Indonesia was officially reassigned to the Western Pacific region following the 78th World Health Assembly[64].

Due to the variability in the definitions of MLTC and the challenges in determining MLTC status across different regions, MLTC eligibility was based on satisfying one or more of the following pre-defined criteria. Firstly, studies were eligible for inclusion based on MLTC status if they were population-based and described the population as multimorbid or having MLTC, comprising of two or more of the conditions outlined in a recent global Delphi consensus study[65]. Secondly, studies investigating a population with a 100% prevalence of an indexed eligible condition and reporting the prevalence of ≥70% of another eligible condition were included. Surveillance studies or equivalent were eligible if at least two eligible conditions were reported in the study sample at a prevalence of ≥70% for each condition. Self-reported conditions were eligible if they met the primary study threshold and satisfied the criteria of ≥70% prevalence. A cut-off of 70% was applied following the definition of a recent scoping review where the MLTC population was defined as over 50% within the sample[66].

Studies from which it was not possible to extract relevant patient characteristics including number and type of conditions, were excluded. Case reports, conference proceedings, posters, and book chapters were excluded. Additionally, reference lists of relevant reviews were hand searched to identify any potential further eligible articles. All references identified via database searching were collected in Covidence systematic review software™. The titles and abstracts in the first stage, and full-text articles in the second stage, were screened by a total of 15 reviewers. Each article was independently screened by at least two reviewers to ensure it met the eligibility criteria. Conflicts at each stage of screening were resolved by a third reviewer or consensus.

The protocol was registered on the International Prospective Register of Systematic Reviews (PROSPERO): CRD42023402674.

### Data extraction

Data was extracted independently by two reviewers using a pre-specified and piloted data extraction form with a third reviewer checking for accuracy. Conflicts arising at this stage were resolved by consensus. Extracted data included: authors, date of publication, sociodemographic data, eligible condition and prevalence, QOL tool, QOL domains and QOL index score. The quality assessment of included studies was conducted using the National Institutes for Health quality assessment tools for observational cohort and cross-sectional studies, pre-post studies with no control group, and controlled interventional studies[67]. Two reviewers independently completed the risk assessment and conflicts were resolved by consensus. An overall rating of good, fair and poor was determined for each study. Studies were not excluded based on their quality assessment rating. A sensitivity analysis was carried out removing studies from the meta-analysis that were scored as poor quality in the risk of bias assessment.

### Data synthesis

Due to heterogeneity in MLTC measurements and definitions, different outcome measures, QOL scores, study designs and missing data, we combined a meta-analysis and narrative synthesis to analyse and present the findings. For the meta-analysis, data was extracted on reported QOL scores from each of the identified studies including the tool used, mean value and standard deviation. Random effects meta-analysis was conducted to pool QOL estimates where the same score was used in three or more study cohorts, and between study heterogeneity was quantified using the I-squared statistic[68]. In one study[29], the WHO-BREF was reported on the 4-20 scale. To enable comparison with other studies using the same tool, this was transformed to the 0–100 scale[69]. All statistical analyses for the meta-analysis were carried out in Stata 18 (Stata Corp, College Station, Texas, USA). Microsoft Excel 365 was used for the narrative synthesis.

For a general description of included studies and the studies reporting the same score in less than three cohorts, a narrative synthesis was conducted following Popay et al.'s iterative approach comprising the following four steps: (1) developing a theoretical model, (2) developing preliminary synthesis, (3) exploring relationships in the data, and 4) assessing the robustness of the synthesis[70].

### Reporting summary

Further information on research design is available in the Nature Portfolio Reporting Summary linked to this article.

## Data availability

The data informing from this review are extracted from previously published studies in the public domain. The dataset generated and analysed during the review is available on Figshare repository (https://doi.org/10.25392/leicester.data.30156316.v1).

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

## Acknowledgements

This research was funded by the NIHR Global Health Research Centre for Multiple Long-Term Conditions using UK aid from the UK Government to support global health research (NIHR203257). The views expressed in this publication are those of the author(s) and not necessarily those of the NIHR or the UK government. K.K. is supported by the National Institute for Health Research (NIHR) Applied Research Collaboration East Midlands (ARC EM), NIHR Global Research Centre for Multiple Long-Term Conditions, NIHR Cross NIHR Collaboration for Multiple Long-Term Conditions, NIHR Leicester Biomedical Research Centre (BRC) and the British Heart Foundation (BHF) Centre of Excellence. M.P.F., D.I., S.C., S.J. and A.P. are supported by the NIHR Applied Research Collaboration East Midlands (ARC EM). P.H. and R.A. are supported by an Advanced Research Fellowship award from the National Institute of Health and Care Research (NIHR303176). S.C. is supported by the supported by the National Institute for Health Research (NIHR) Global Health Research Group for Cardiometabolic Disease Research in Africa Partnership (CREATE) (NIHR132995).

## Author contributions

P.H., S.M., N.S.V., D.P. and K.K. conceptualised the study. D.I., P.H., R.A., A.A., A.B.A., A.B.R., S.C., M.P.F., S.J., P.M., and A.P. performed data curation. C.G. and D.I. carried out data analysis and validation of the data. D.I. wrote the first draft of the manuscript. D.I., P.H., C.G., M.A., A.B.A., S.C., M.P.F., S.G., D.K., S.M., N.O., K.S., A.V., N.S.V., D.P. and K.K. reviewed and edited the manuscript. All authors approved submission of the final manuscript for publication.

## Competing interests

K.K. has acted as a consultant, speaker or received grants for investigator-initiated studies for Astra Zeneca, Bayer, Novo Nordisk, Sanofi-Aventis, Servier, Lilly and Merck Sharp & Dohme, Boehringer Ingelheim, Oramed Pharmaceuticals, Pfizer, Roche, Daiichi-Sankyo, Applied Therapeutics, Embecta and Nestle Health Science. The remaining authors declare no competing interests.
