## [Transparent Peer Review file · Nature Communications]

Quality of life and multiple long-term conditions in Southeast Asia: a systematic review and meta-analysis

Corresponding Author: Dr Deborah Ikhile

Version 0:

Reviewer comments:

Reviewer #2

(Remarks to the Author)

This is an important and timely article addressing quality of life measurement among individuals living with multiple long-term conditions in Southeast Asia. The study is notable for its comprehensive inclusion of participants and its use of several established and validated quality of life assessment tools. As a clinician, I find the focus on MLTCs particularly relevant, as many of our patients fall into this category, and the findings of this study could meaningfully contribute to the existing literature.

I have only a few minor comments:

1. **Regional Representation:** While the study includes a substantial number of studies from South Asia—particularly India, Nepal, Bangladesh, and Sri Lanka—only a small number (n=12) are from Southeast Asia. Several countries such as Myanmar and Laos appear to be missing. It would be helpful if the authors could clarify how Southeast Asia was defined for the purpose of this review (e.g., WHO regional classification vs. other frameworks).
2. **Disease Categorization:** It may strengthen the paper to discuss metabolic dysfunction-associated steatotic liver disease, which are prevalent in the region and significantly impact quality of life, as part of the broader context of MLTCs.
3. **Prevalence Underestimation:** The reported burden of MLTCs may be underestimated due to potential gaps in diagnosis or reporting. Given the high prevalence of many chronic conditions in the region, this deserves acknowledgment.
4. **Policy Implications:** The use of validated tools in this study is a valuable strength that enhances its robustness. The authors may wish to elaborate on the broader implications of their findings—particularly how MLTCs contribute to long-term disability and the increased demand on healthcare systems in aging Southeast Asian populations.

Reviewer #3

(Remarks to the Author)

This review aims to systematically synthesise evidence on quality of life (QOL) measures and their associated factors among individuals living with multiple long-term conditions (MLTCs) in Southeast Asia. The authors included 34 studies in the narrative synthesis and 14 studies in the meta-analysis. The topic is timely and relevant, particularly given the growing burden of MLTCs in the region and the lack of region-specific syntheses on QOL outcomes. It highlights the heterogeneity in QOL instruments and study designs across the region. However, the current presentation of findings — particularly the handling of the meta-analysis and narrative synthesis — weakens the potential contribution.

1. The forest plot (Figure 3) presented do not include combined effect sizes or heterogeneity statistics, which are essential features of meta-analysis.
2. While different QOL instruments were used across studies, standardising scores (e.g., transforming to a 0–100 scale) could have allowed for more inclusive meta-analyses. Was this considered?
3. The author focus on pooling estimates by type of QOL measures. It would be more valuable if we can look at the patterns

in QOL measures by disease combinations.

4. The inclusion of low-quality studies without sensitivity analysis undermines confidence in some of the conclusions

5. Several factors — such as age, gender, and education level — were reported in multiple studies. It is unclear why a meta-analysis was not conducted for these factors when they were reported in at least three studies. The authors should consider pooling effect sizes where possible. Additionally, including the effect sizes and corresponding sample sizes in the narrative synthesis would strengthen the interpretation and allow for clearer comparisons across studies.

Version 1:

Reviewer comments:

Reviewer #2

(Remarks to the Author)

Author already did an excellent work revising the manuscript

Reviewer #3

(Remarks to the Author)

The authors have addressed all my comments as well as those of the other reviewers. I consider the paper now suitable for publication.

REVIEWER COMMENTS

We would like to thank the reviewers for the time and effort dedicated to reviewing our manuscript. We have given point by point responses to all the reviewers comments and have made the suggested changes to the manuscript. We hope these changes satisfactorily address the reviewers' feedback.

Reviewer #2 (Remarks to the Author):

This is an important and timely article addressing quality of life measurement among individuals living with multiple long-term conditions in Southeast Asia. The study is notable for its comprehensive inclusion of participants and its use of several established and validated quality of life assessment tools. As a clinician, I find the focus on MLTCs particularly relevant, as many of our patients fall into this category, and the findings of this study could meaningfully contribute to the existing literature.

We appreciate the reviewer's comments and encouraging feedback. We are delighted to hear that the article resonated with you, particularly from a clinical perspective.

I have only a few minor comments:

1. Regional Representation: While the study includes a substantial number of studies from South Asia—particularly India, Nepal, Bangladesh, and Sri Lanka—only a small number (n=12) are from Southeast Asia. Several countries such as Myanmar and Laos appear to be missing. It would be helpful if the authors could clarify how Southeast Asia was defined for the purpose of this review (e.g., WHO regional classification vs. other frameworks).

We thank the reviewer for highlighting this. We used the 11 countries from the WHO South-East Asia region classification for our review. We have revised our methods section on page 10 to clarify as follows:

...as defined by the World Health Organization South-East Asia as of August 2024⁶³, were included. These included Bangladesh, Bhutan, Democratic People's Republic of Korea, India, Indonesia, Maldives, Myanmar (Burma), Nepal, Sri Lanka, Thailand, and Timor-Leste (East Timor, Democratic Republic of Timor-Leste). It is important to note that Indonesia was officially reassigned to the Western Pacific region following the 78th World Health Assembly⁶⁴.

2. Disease Categorization: It may strengthen the paper to discuss metabolic dysfunction-associated steatotic liver disease, which are prevalent in the region and significantly impact quality of life, as part of the broader context of MLTCs.

We thank the reviewer for this observation. While we agree that metabolic dysfunction-associated steatotic liver disease (MASLD) is an important and prevalent condition in Southeast Asia, however, we looked through the available evidence and could not find any QOL publications on MASLD in the context of MLTC.

Also, since the scope of our review was not to assess or analyse the impact of specific conditions in isolation, we therefore feel it is appropriate to retain our discussion at the level of MLTC generally, rather than on condition-specific analyses.

3. Prevalence Underestimation: The reported burden of MLTCs may be underestimated due to potential gaps in diagnosis or reporting. Given the high prevalence of many chronic conditions in the region, this deserves acknowledgment.

We agree with this comment, and we have included the following sentence in the introduction (page 2) to acknowledge the likely underestimation of MLTC burden in the region:

It is crucial to note that estimated burden of MLTC in the region, as is the case in other Low-and-Middle-income Countries LMICs is likely underestimated due to suboptimal reporting and poorly integrated health systems².

We have also acknowledged this in the discussion (page 9):

Firstly, the underestimation of MLTC burden in the region leads to an incomplete assessment of QOL outcomes in people living with MLTC.

4. Policy Implications: The use of validated tools in this study is a valuable strength that enhances its robustness. The authors may wish to elaborate on the broader implications of their findings—particularly how MLTCs contribute to long-term disability and the increased demand on healthcare systems in aging Southeast Asian populations.

We thank the reviewer for this comment. We agree and we have included the below paragraph (page 9) highlighting the policy implications of our finding in relation to how MLTC and long-term disability:

Our review also has significant policy implications, particularly in relation to QOL and long-term disability management in resource-constrained healthcare systems. Although limited data exists for the Southeast Asia region, a meta-analysis published in 2024 estimated a high prevalence of disability among people with MLTC at 34.9% (95% CI = 25.8-43.9%)⁶⁰. The effect of MLTC on long-term disability exacerbates healthcare utilisation and expenditure, increases caregiving burden, and the need for rehabilitation and long-term care, which many health systems in the region are not yet well-equipped to provide. Policy responses need to include patient centric approaches rather than disease specific approaches, rather than disease specific vertical approaches, investment in universal health coverage as a way of addressing disparities due to the social determinants of health and promoting preventive efforts, particularly those targeting modifiable risk factors. Also, investment in quality curative and rehabilitation services is required to mitigate adverse impact on QOL due to disability resulting from MLTC.

Reviewer #3 (Remarks to the Author):

This review aims to systematically synthesise evidence on quality of life (QOL) measures and their associated factors among individuals living with multiple long-term conditions (MLTCs) in Southeast Asia. The authors included 34 studies in the narrative synthesis and 14 studies in the meta-analysis. The topic is timely and relevant, particularly given the growing burden of MLTCs in the region and

the lack of region-specific syntheses on QOL outcomes. It highlights the heterogeneity in QOL instruments and study designs across the region. However, the current presentation of findings — particularly the handling of the meta-analysis and narrative synthesis — weakens the potential contribution.

We appreciate the reviewer's comments and we have carefully considered all their feedback, providing point-by-point responses.

1. The forest plot (Figure 3) presented do not include combined effect sizes or heterogeneity statistics, which are essential features of meta-analysis.

We have now updated the forest plot to include the pooled estimates (Figure 2).

2. While different QOL instruments were used across studies, standardising scores (e.g., transforming to a 0–100 scale) could have allowed for more inclusive meta-analyses. Was this considered?

We thank the reviewer for this suggestion. This was considered but due to substantial heterogeneity between QoL scores (e.g., some were generic and some were disease specific), we decided to pool results for each tool separately. This follows good practice guidance (e.g., according to the Cochrane Handbook (Higgins et al., 2024) for meta-analyses which recommends not pooling result outcomes that are considered too heterogeneous. We believe this allows our results to be more easily interpreted.

3. The author focus on pooling estimates by type of QOL measures. It would be more valuable if we can look at the patterns in QOL measures by disease combinations.

We agree this would be of interest but as we only had aggregated data for each study we were unable to explore this.

4. The inclusion of low-quality studies without sensitivity analysis undermines confidence in some of the conclusions

Thank you. This is a valid point and we agree with the reviewer's comment. On the reviewer's suggestion, we have added in a sensitivity analysis removing the studies classified as poor quality. We have revised the results and methods sections accordingly:

Results (page 4): A sensitivity analyses excluding one of these studies resulted in a lower pooled EQ-VAS mean score of 58.43 (56.35, 60.51). The second poorly rated study utilised the Parkinson's disease questionnaire, but as this was the only study using this tool a sensitivity analysis for study quality could not be carried out.

Methods (page 11): A sensitivity analysis was carried out removing studies from the meta-analysis that were scored as poor quality in the risk of bias assessment.

5. Several factors — such as age, gender, and education level — were reported in multiple studies. It is unclear why a meta-analysis was not conducted for these factors when they were reported in at least three studies. The authors should consider pooling effect sizes where possible. Additionally, including the effect sizes and corresponding sample sizes in the narrative synthesis would strengthen

the interpretation and allow for clearer comparisons across studies.

Meta-analysis is primarily used to pool outcomes across studies rather than patient demographics. The main aim of this work was to assess QoL (tools utilised and scores) in MLTC cohorts, and as such we focused our analyses on this.

Once again, we sincerely thank both reviewers for the time spent reviewing our work and their comments.